# LEARNING MANIFOLD PATCH-BASED REPRESENTATIONS OF MAN-MADE SHAPES

**Dmitriy Smirnov**
MIT
smirnov@mit.edu

**Mikhail Bessmeltsev**
Université de Montréal
bmpix@iro.umontreal.ca

**Justin Solomon**
MIT
jsolomon@mit.edu

## ABSTRACT

Choosing the right representation for geometry is crucial for making 3D models compatible with existing applications. Focusing on piecewise-smooth man-made shapes, we propose a representation that is usable in conventional CAD modeling pipelines and can also be learned by deep neural networks. We demonstrate its benefits by applying it to the task of sketch-based modeling. Given a raster image, our system infers a set of parametric surfaces that realize the input in 3D. To capture piecewise smooth geometry, we learn a special shape representation: a deformable parametric template composed of Coons patches. Naïvely training such a system, however, is hampered by non-manifold artifacts in the parametric shapes and by a lack of data. To address this, we introduce loss functions that bias the network to output non-self-intersecting shapes and implement them as part of a fully self-supervised system, automatically generating both shape templates and synthetic training data. We develop a testbed for sketch-based modeling, demonstrate shape interpolation, and provide comparison to related work.

## 1 INTRODUCTION

While state-of-the art deep learning systems that output 3D geometry as point clouds, triangle meshes, voxel grids, and implicit surfaces yield detailed results, these representations are dense, high-dimensional, and incompatible with CAD modeling pipelines. In this work, we develop a 3D representation that is parsimonious, geometrically interpretable, and easily editable with standard tools, while being compatible with deep learning. This enables a shape modeling system leveraging the ability of neural networks to process incomplete, ambiguous input and produces useful, consistent 3D output.

Our primary technical contributions involve the development of machinery for learning *parametric* 3D surfaces in a fashion that is efficiently compatible with modern deep learning pipelines and effective for a challenging 3D modeling task. We automatically infer a template per shape category and incorporate loss functions that operate explicitly on the geometry rather than in the parametric domain or on a sampling of surrounding space. Extending learning methodologies from images and point sets to more exotic modalities like networks of surface patches is a central theme of modern graphics, vision, and learning research, and we anticipate broad application of these developments in CAD workflows.

To test our system, we choose sketch-based modeling as a target application. Converting rough, incomplete 2D input into a clean, complete 3D shape is extremely ill-posed, requiring hallucination of missing parts and interpretation of noisy signal. To cope with these ambiguities, existing systems either rely on hand-designed priors, severely limiting applications, or learn the shapes from data, implicitly inferring relevant priors (Delanoy et al., 2018; Wang et al., 2018a; Lun et al., 2017). However, the output of the latter methods often lacks resolution and sharp features necessary for high-quality 3D modeling.

In industrial design, man-made shapes are typically modeled as collections of smooth parametric patches (e.g., NURBS surfaces) whose boundaries form the sharp features. To learn such shapes effectively, we use a deformable parametric template (Jain et al., 1998)—a manifold surface composed of patches, each parameterized by control points (Fig. 3a). This representation enables the model to control the smoothness of each patch and introduce sharp edges between patches where necessary.

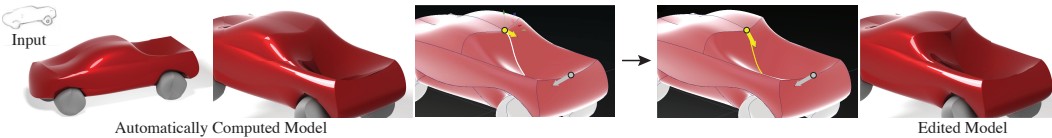

Figure 1: Editing a 3D model produced by our method. Because we output 3D geometry as a collection of consistent, well-placed NURBS patches, user edits can be made in conventional CAD software by simply moving control points. Here, we are able to refine the trunk of a car model with just a few clicks.

Compared to traditional representations, deformable parametric templates have numerous benefits for our task. They are intuitive to edit with conventional software, are resolution-independent, and can be meshed to arbitrary accuracy. Since only boundary control points are needed, our representation has relatively few parameters. Finally, this structure admits closed-form expressions for normals and other geometric features, which can be used for loss functions that improve reconstruction quality (§3.2).

Training a model for such representations faces three major challenges: detection of non-manifold surfaces, structural variation within shape categories, and lack of data. We address them as follows:

- We introduce several loss functions that encourage our patch-based output to form a manifold mesh without topological artifacts or self-intersections.
- Some categories of man-made shapes exhibit structural variation. To address this, for each category we algorithmically generate a varying deformable template, which allows us to support separate structural variation using a variable number of parts (§3.1).
- Supervised methods mapping from sketches to 3D require a database of sketch-model pairs, and, to-date, there are no such large-scale repositories. We use a synthetic sketch augmentation pipeline inspired by artistic literature to simulate variations observed in natural drawings (§4.1). Although our model is trained on synthetic sketches, it generalizes to natural sketches.

Our method is self-supervised: We predict patch parameters, but our data is not labeled with patches.

## 2 RELATED WORK

### 2.1 DEEP LEARNING FOR SHAPE RECONSTRUCTION

Learning to reconstruct 3D geometry has recently enjoyed significant research interest. Typical forms of input are images (Gao et al., 2019; Wu et al., 2017; Delanoy et al., 2018; Häne et al., 2019) and point clouds (Williams et al., 2019; Groueix et al., 2018; Park et al., 2019). When designing a network for this task, two considerations affect the architecture: the loss function and the geometric representation.

**Loss Functions.** One popular direction employs a differentiable renderer and measures 2D image loss between a rendering of the inferred 3D model and the input image (Kato et al., 2018; Wu et al., 2016; Yan et al., 2016; Rezende et al., 2016; Wu et al., 2017; Tulsiani et al., 2017b; 2018). A notable example is the work by Wu et al. (2017), which learns a mapping from a photograph to a normal map, a depth map, a silhouette, and the mapping from these outputs to a voxelization. They use a differentiable renderer and measure inconsistencies in 2D. Hand-drawn sketches, however, cannot be interpreted as perfect projections of 3D objects: They are imprecise and often inconsistent (Bessmeltsev et al., 2016). Another approach uses 3D loss functions, measuring discrepancies between the predicted and target 3D shapes directly, often via Chamfer or a regularized Wasserstein distance (Williams et al., 2019; Liu et al., 2010; Mandikal et al., 2018; Groueix et al., 2018; Park et al., 2019; Gao et al., 2019; Häne et al., 2019). We build on this work, adapting Chamfer distance to patch-based geometric representations and extending the loss function with new regularizers (§3.2).

**Shape representation.** As noted by Park et al. (2019), geometric representations in deep learning broadly can be divided into three classes: voxel-, point-, and mesh-based. Voxel-based methods (Delanoy et al., 2018; Wu et al., 2017; Zhang et al., 2018; Wu et al., 2018) yield dense reconstruction that are limited in resolution, offer no topological guarantees, and cannot represent sharp features. Point-based approaches represent geometry as a point cloud (Yin et al., 2018; Mandikal et al., 2018; Fan et al., 2017; Lun et al., 2017; Yang et al., 2018), sidestepping memory issues, but do not capture manifold connectivity.

Some recent methods represent shapes using meshes (Bagautdinov et al., 2018; Baque et al., 2018; Litany et al., 2018; Kanazawa et al., 2018; Wang et al., 2019; Nash et al., 2020). Our parametric template representation allows us to more easily enforce piecewise smoothness and test for self-intersections (§3.2). These properties are difficult to measure on meshes in a differentiable manner. Compared to a generic template shape, such as a sphere, our *category-specific* templates improve reconstruction quality and enable complex reconstruction constraints, e.g., symmetry. We compare to the deformable mesh representations in §4.5.

Most importantly, our shape representation is *native* to modern CAD software and can be directly edited in this software, as demonstrated in Figure 1. The key to this flexibility is the type of the parametric patches we use, which can be trivially converted to a NURBS representation (Piegl & Tiller, 1996), the standard surface type in CAD. Other common shape representations, such as meshes or point clouds, cannot be easily converted into NURBS format: algorithmically fitting NURBS surfaces is nontrivial and is an active area of research (Yumer & Kara, 2012; Krishnamurthy & Levoy, 1996).

Finally, some recent works parameterize 3D geometry using learned deep neural networks—e.g., they learn an implicit representation (Mescheder et al., 2019; Chen & Zhang, 2019; Genova et al., 2019) or a mapping from parameter space to a collection of patches (Groueix et al., 2018; Deng et al., 2020). These demonstrate impressive results but are not tuned to CAD applications; it is unclear how their output can be converted to editable CAD shape representations.

## 2.2 Sketch-based 3D shape modeling

3D reconstruction from sketches has a long history in graphics. A survey is beyond the scope of this paper; see (Delanoy et al., 2018) or surveys by Ding & Liu (2016) and Olsen et al. (2009).

Unlike incremental sketch-based 3D modeling, where users progressively add new strokes (Cherlin et al., 2005; Gingold et al., 2009; Chen et al., 2013; Igarashi et al., 1999), our method interprets complete sketches, eliminating training for artists and enabling 3D reconstruction of legacy sketches.

Some systems interpret complete sketches without extra information. This input is extremely ambiguous thanks to occlusions and inaccuracies. Hence, reconstruction algorithms rely on strong 3D shape priors. These priors are typically manually created, e.g., for humanoids, animals, and natural shapes (Bessmeltsev et al., 2015; Entem et al., 2015; Igarashi et al., 1999). Our work focuses on man-made shapes, which have characteristic sharp edges and are only piecewise smooth. Rather than relying on expert-designed priors, we *automatically* learn category-specific shape priors.

A few deep learning approaches address sketch-based modeling. Nishida et al. (2016) and Huang et al. (2017) train networks to predict *procedural model parameters* that yield detailed shapes from a sketch. These methods produce complex high-resolution models but only for shapes that can be procedurally generated. Lun et al. (2017) use a CNN-based architecture to predict multi-view depth and normal maps, later converted to point clouds; Li et al. (2018) improve on their results by first predicting a flow field from an annotated sketch. In contrast, we output a deformable parametric template, which can be converted to a manifold mesh without post-processing. Wang et al. (2018a) learn from unlabeled databases of sketches and 3D models with no correspondence using an adverserial training approach. Another inspiration for our research is the work of Delanoy et al. (2018), which reconstructs a 3D object as voxel grids; we compare to this work in Fig. 7.

## 3 Algorithm

Our learning pipeline outputs a parametrically-defined 3D surface. We describe our geometric representation (§3.1), define our loss (§3.2), and specify our architecture and training procedure (§3.3).

### 3.1 Representation

**Patches.** Our surfaces are collections of *Coons patches* (Coons, 1967), a commonly used and rich subset of NURBS surfaces. A patch is specified by four boundary cubic Bézier curves sharing endpoints (Fig. 3a). Each curve has control points $p_1, ..., p_4 \in \mathbb{R}^3$. A Bézier curve $c : [0,1] \to \mathbb{R}^3$ is

Figure 3: Our representation is composed of Coons patches (a) organized into a deformable template (b). We use the following templates (c, top to bottom, left to right): bottle, knife, guitar, car, airplane, coffee mug, gun, bathtub, 24-patch sphere, 54-patch sphere.

$$c(\gamma)\!=\!p_1(1\!-\!\gamma)^3\!+\!3p_2\gamma(1\!-\!\gamma)^2\!+\!3p_3\gamma^2(1\!-\!\gamma)\!+\!p_4\gamma^3, \text{ and a Coons patch } P\!:\![0,1]^2\!\rightarrow\!\mathbb{R}^3 \text{ is}$$
$$P(s,t)\!=\!(1\!-\!t)c_1(s)\!+\!tc_3(1\!-\!s)\!+\!sc_2(t)\!+\!(1\!-\!s)c_4(1\!-\!t)$$
$$-(c_1(0)(1\!-\!s)(1\!-\!t)\!+\!c_1(1)s(1\!-\!t)\!+\!c_3(1)(1\!-\!s)t\!+\!c_3(0))\,st. \quad (1)$$

**Templates.** Templates specify patch connectivity. A template consists of the minimal number of control points necessary to define the patches; shared control points are reused (Fig. 3b, c). We allow the edge of one patch to be contained *within* the edge of another using *junction curves*. A *junction curve* $c^d$ is constrained to a lie along a parent curve $d$ and is thus parameterized by $s,t\in[0,1]$, such that $c(0)\!=\!d(s)$ and $c(1)\!=\!d(t)$.

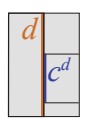

A template provides hard topological constraints for our surfaces, an initialization of their geometry, and, optionally, a means for regularization. Templates are crucial in ensuring that the patches have consistent topology—an approach without templates would result in unstructured, non-manifold patch collections. While our method works using a generic sphere template, we can define templates per shape category to incorporate category-specific priors. These templates capture only coarse geometric features and approximate scale. We outline an algorithm for obtaining templates below.

**Algorithmic construction of templates.** We design a simple system to construct a template automatically given as input any collection of cuboids. Such a collection can be computed automatically for a shape category, e.g., given a segmentation or using self-supervised methods such as (Smirnov et al., 2020; Tulsiani et al., 2017a; Sun et al., 2019), or easily produced using standard CAD software. We show templates algorithmically computed from pre-segmented shapes—we obtain a collection of cuboids by taking the bounding box around each connected component of each segmentation class.

A generic cuboid decomposition cannot be used as a template, since individual cuboids may overlap. We snap cuboids to an integer lattice, split each face at grid coordinates, and remove overlapping and interior faces to obtain a manifold quad mesh. This mesh typically consists of many faces, and so, we simplify it. We merge adjacent quads with a greedy agglomerative algorithm, iterating over each quad in order of descending area and merging with an adjacent quad as long as the merge does not result in ill-defined junction

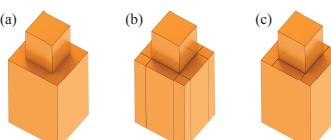

Figure 2: Summary of our template algorithm. Given a collection of cuboids (a), we form a quad mesh (b), and merge faces to get a template (c).

curves. We show an example of this process in Fig. 2. Given decompositions of multiple shapes in a category, we use the median model w.r.t. Chamfer distance. Since models within a category are aligned, the median provides a rough approximation of the typical geometry.

**Structural variation using templates.** For category-specific templates, we use the fact that template patches are consistently placed on semantically meaningful parts to account for structural variation.

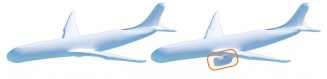

For instance, certain airplane models contain turbines, while others do not. We note which template patches come from cuboids corresponding to turbines, and, during training, only use turbine patches for models that contain turbines. This allows training on the entire airplane shape category, effectively using two distinct templates. At test time, the user can toggle turbines on or off for any output model.

## 3.2 Loss

We fit a collection of patches $\{P_i\}$ to a target mesh $M$ by optimizing a differentiable loss function. Below, we describe each term—a reconstruction loss generalizing Chamfer distance, a normal alignment loss, a self-intersection regularizer, a patch flatness regularizer, and two template-based priors.

**Area-weighted Chamfer distance.** Given two measurable shapes $A, B \subset \mathbb{R}^3$ and point sets $X$ and $Y$ sampled from $A$ and $B$, respectively, the *directed Chamfer distance* between $X$ and $Y$ is

$$\text{Ch}_{\text{dir}}(X, Y) = \frac{1}{|X|} \sum_{x \in X} \min_{y \in Y} \text{d}(x, y), \tag{2}$$

where $\text{d}(x, y) = \|x - y\|_2$. The *symmetric Chamfer distance* is $\text{Ch}(X, Y) = \text{Ch}_{\text{dir}}(X, Y) + \text{Ch}_{\text{dir}}(Y, X)$.

Chamfer distance is differentiable and thus a popular loss function in deep learning. However, the distribution under which $X$ and $Y$ are sampled has significant impact. Sampling uniformly in patch parameter space (a unit square) does not capture the surface area measure and, consequently, does not yield uniform samples on the surface—regions with higher curvature end up oversampled. A closed form arc-length parameterization does not even exist for a Bézier curve, and so a uniform parameterization for even a one-dimensional patch is difficult to compute. Many works try to address this issue by designing custom sampling procedures (Wang et al., 2012; Hernández-Mederos & Estrada-Sarlabous, 2003; Elkott et al., 2002). These schemes are typically computationally expensive and/or incompatible with a differentiable learning pipeline. To address this issue, following Smirnov et al. (2020), we first define *variational directed Chamfer distance*, starting from equation 2:

$$\text{Ch}_{\text{dir}}(X, Y) = \frac{1}{|X|} \sum_{x \in X} \min_{y \in Y} \text{d}(x, y) \approx \mathbb{E}_{x \sim \mathcal{U}_A} \left[ \inf_{y \in Y} \text{d}(x, y) \right] = \frac{1}{\text{Area}(X)} \int_X \inf_{y \in Y} \text{d}(x, y) \text{d}x \overset{\text{def}}{=} \text{Ch}_{\text{dir}}^{\text{var}}(A, B), \tag{3}$$

where $\mathcal{U}_A$ is the uniform distribution on $A$. $\text{Ch}^{\text{var}}(A, B)$ is defined analogously.

While it is difficult to sample uniformly from patches, we can do so easily from their parametric domain (the unit square). We perform a change of variables to get the following (derivation in §A.1):

$$\text{Ch}_{\text{dir}}^{\text{var}}(P, M) = \frac{\mathbb{E}_{(s,t) \sim \mathcal{U}_{[0,1]^2}}[\inf_{y \in M} \text{d}(P(s,t), y) |J(s,t)|]}{\mathbb{E}_{(s,t) \sim \mathcal{U}_{[0,1]^2}}[|J(s,t)|]}, \tag{4}$$

where $J(s, t)$ is the Jacobian of $P(s, t)$. We approximate this expression via Monte Carlo integration.

Since we can precompute uniformly sampled random points from the target mesh, we do not need to use area weights for $\text{Ch}_{\text{dir}}^{\text{var}}(M, P)$. Thus, our area-weighted Chamfer distance is

$$\mathcal{L}_{\text{Ch}}(\cup P_i, M) = \frac{\sum_i \sum_{(s,t) \in U_\square} \min_{y \in M} \text{d}(P(s,t), y)|J_i(s,t)|}{\sum_i \sum_{(s,t) \in U_\square} |J_i(s,t)|} + \frac{\sum_{x \in M} \min_{y \in \cup P_i} \text{d}(x, y)}{|M|}, \tag{5}$$

where $U_\square \sim \mathcal{U}_{[0,1]^2}$. We compute $J_i(u, v)$ for a patch given its control points in closed-form.

**Normal alignment.** The normal alignment term is computed analogously to $\text{Ch}_{\text{dir}}$, except that instead of Euclidean distance, we use $\text{d}_N(x, y) = \|n_x - n_y\|_2^2$, where $n_x$ is the unit normal at $x$. For each point $y$ sampled from our predicted surface, we compare $n_y$ to $n_x$, where $x \in M$ is closest to $y$, and, symmetrically, for each $x' \in M$, we compare $n_{x'}$ to to $n_{y'}$, where $y' \in \cup P_i$ is closest to $x'$:

$$\mathcal{L}_{\text{normal}}(\cup P_i, M) = \frac{\sum_i \sum_{(u,v) \in U_\square} \text{d}_N(\text{NN}_M(P_i(u,v)), P_i(u,v))|J_i(u,v)|}{\sum_i \sum_{(u,v) \in U_\square} |J_i(u,v)|} + \frac{\sum_{x \in M} \text{d}_N(x, \text{NN}_{\cup P_i}(x))}{|M|}, \tag{6}$$

where $\text{NN}_Y(x)$ is the nearest neighbor to $x$ in $Y$ under Euclidean distance.

**Intersection regularization.** We introduce a *collision detection loss* to detect patch intersections:

$$\mathcal{L}_{\text{coll}}(\{P_i\}) = \sum_{i \neq j} \exp\left(-\left(\min(\text{d}(\mathcal{T}^i, P^j), \text{d}(\mathcal{T}^j, P^i))/\varepsilon\right)^2\right), \tag{7}$$

where $\mathcal{T}^i$ is a triangulation of patch $P_i$, and $P^i$ is a set of points sampled from patch $P_i$. To triangulate a patch, we take a fixed triangulation of the parameter space (a unit square) and compute the image of each vertex under the Coons patch map, keeping the original connectivity. With a small $\varepsilon = 10^{-6}$, this expression is a smooth indicator for when two patches are intersecting. For a pair of adjacent patches or those that share a junction, we truncate one patch at the adjacency before evaluating the loss.

**Patch flatness regularization.** To help patches to align to smooth regions and sharp creases to fall on patch boundaries, we define a *patch flatness regularizer*, which discourages excessive curvature by regularizing each Coons patch map $P : [0,1] \times [0,1] \to \mathbb{R}^3$ to be close to a linear map. For each patch, we sample points $U_\square$ in parameter space, compute their image $P(U_\square)$, and fit a linear function using

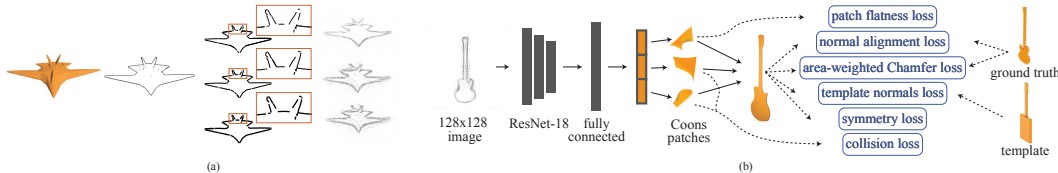

Figure 4: An overview of our data generation and augmentation (a) and learning (b) pipelines.

least-squares. Thus, $\hat{P}(U_\square) = AU_\square + b \approx P(U_\square)$ for some $A, b$. We define patch flatness loss as

$$\mathcal{L}_{\text{flat}}(\{P_i\}) = \frac{\sum_i \sum_{(u,v) \in U_\square} \|\hat{P}_i(u,v) - P_i(u,v)\|_2^2 |J_i(u,v)|}{\sum_i \sum_{(u,v) \in U_\square} |J_i(u,v)|}. \tag{8}$$

**Template normals regularization.** For categories where a template is available, we not only initialize the network with the template geometry but also regularize the output using template normals. This favorably positions patch seams and prevents patches from sliding over high-curvature regions:

$$\mathcal{L}_{\text{template}}(\{P_i\}, \{T_i\}) = \frac{\sum_i \sum_{(u,v) \in U_\square} \|n_{P_i(u,v)} - n_{T_i}\|_2^2 |J_i(u,v)|}{\sum_i \sum_{(u,v) \in U_\square} |J_i(u,v)|}, \tag{9}$$

where $n_{T_i}$ is the normal vector of the $i$th template patch.

**Global symmetry.** Man-made shapes often exhibit global bilateral symmetries. Enforcing symmetry is difficult in, e.g., meshes or implicit surfaces. In contrast, after computing a template's symmetry planes, we enforce symmetric positions of the corresponding control points as an additional loss term:

$$\mathcal{L}_{\text{sym}}(\cup P_i) = \frac{1}{|S|} \sum_{(i,j) \in S} \|(P_x^i - a, P_y^i, P_z^i) - (a - P_x^j, P_y^j, P_z^j)\|_2^2, \tag{10}$$

where $S$ contains index pairs of symmetric control points $(P^i, P^j)$. In the formula, we assume w.l.o.g. symmetry plane $x = a$. We use this to enforce symmetrical reconstruction of airplanes and cars.

### 3.3 DEEP LEARNING PIPELINE

The final loss that we optimize is

$$\begin{aligned}\mathcal{L}(\{P_i\}, M) = {} & \mathcal{L}_{\text{Ch}}(\cup P_i M) + \alpha_{\text{normal}} \mathcal{L}_{\text{normal}}(\cup P_i M) \\ & + \alpha_{\text{flat}} \mathcal{L}_{\text{flat}}(\{P_i\}) + \alpha_{\text{coll}} \mathcal{L}_{\text{coll}}(\{P_i\}) + \alpha_{\text{template}} \mathcal{L}_{\text{template}}(\{P_i\}, \{T_i\}) + \alpha_{\text{sym}} \mathcal{L}_{\text{sym}}(\cup P_i).\end{aligned} \tag{11}$$

For models scaled to fit in a unit sphere, we use $\alpha_{\text{normal}} = 0.008$, $\alpha_{\text{flat}} = 2$, and $\alpha_{\text{coll}} = 0.00001$ for all experiments, and $\alpha_{\text{template}} = 0.0001$ and $\alpha_{\text{sym}} = 1$ for experiments that use those regularizers.

Our network inputs one or more $128 \times 128$ images and outputs patch parameters. The architecture is ResNet-18 (He et al., 2016) followed by hidden layers with 1024, 512, and 256 units, and an output layer with size equal to the output dimension. Final layer weights are initialized to zero with bias equal to the template parameters. For multi-view input, we encode each image and do max pooling over the latent codes. We use ReLU and batch normalization after each layer except the last. We train each network for 24 hours on a Tesla V100 GPU, using Adam (Kingma & Ba, 2014) and batch size 8 with learning rate 0.0001. At each iteration, we sample 7,000 points from the predicted and target shapes. Our pipeline is illustrated in Fig. 4b.

## 4 EXPERIMENTAL RESULTS

We introduce a pipeline for generating realistic sketch data and train a network that coverts a sketch image to a patch-based 3D representation. We show results on synthetic and human-drawn sketches, demonstrate interpolation and quad meshing, compare to prior work, and do an ablation study (§A.2).

### 4.1 DATA PREPARATION

While there exist annotated datasets of 3D models and corresponding hand-drawn sketches (Gryaditskaya et al., 2019), such data are unavailable at the deep learning scale. Instead, we generate synthetic data. Guided by Cole et al. (2012), we first render occluding contours and sharp edges using the Arnold Toon Shader in Autodesk Maya for each model from representative camera views. Although the contour images capture the main features of the 3D model, they lack some of the ambiguities of

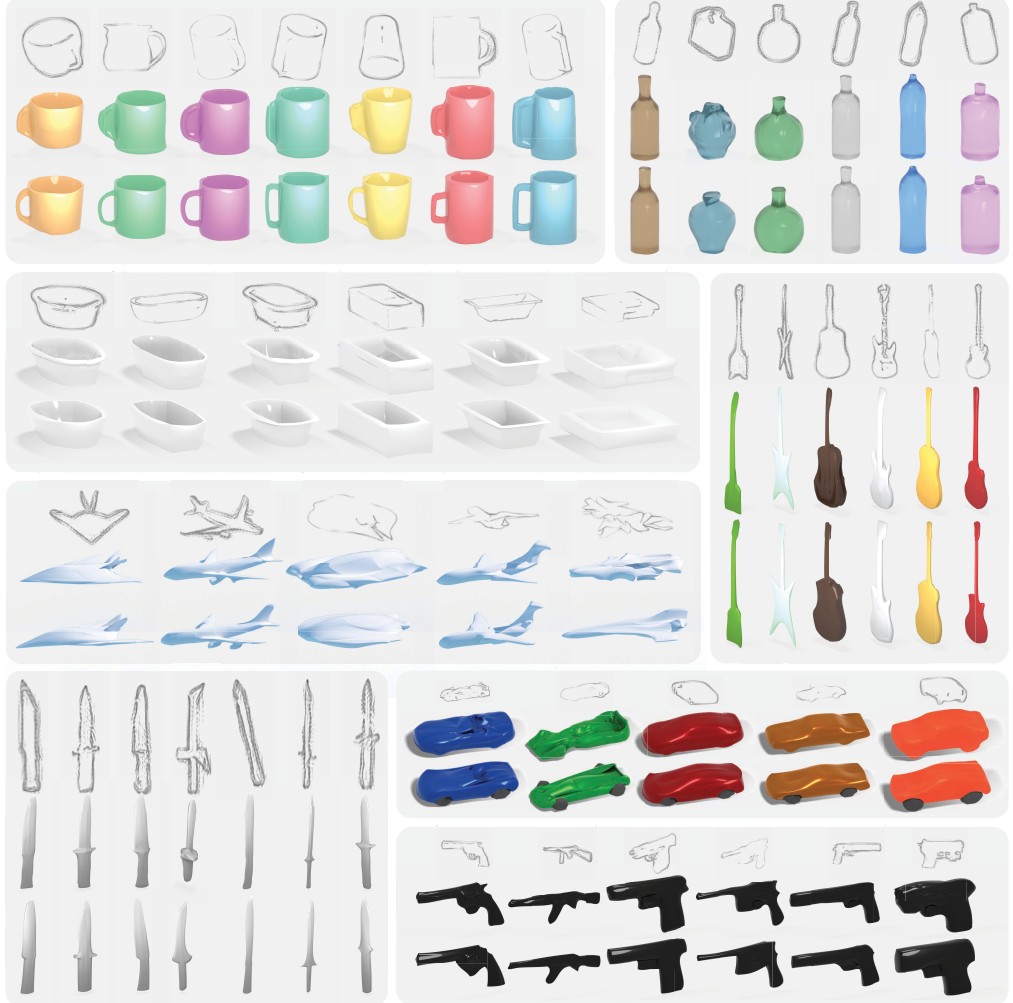

Figure 5: Results on synthetic sketches. For each category, from top to bottom: input sketch, output 3D model with sphere template (54 patches), output 3D model with category-specific template.

rough hand-drawn sketches (Liu et al., 2018). To this end, we vectorize the images using Bessmeltsev & Solomon (2019) and augment the set of contours by stochastically splitting or truncating curves. Finally, we rasterize each contour image using different stroke widths and pass it through the pencil drawing generation model of Simo-Serra et al. (2018). We illustrate this pipeline in Fig. 4a.

Thus, we obtain realistic sketch images paired with 3D models. We train on the airplane, bathtub, guitar, bottle, car, mug, gun, and knife categories of ShapeNet Core (v2) (Chang et al., 2015). We make the meshes watertight using Huang et al. (2018) and normalize them to fit an origin-centered unit sphere.

## 4.2 REAL AND SYNTHETIC SKETCH RECONSTRUCTION

We pick a random 10%-90% test-train split for each category and evaluate in Fig. 5 as well as §A.5.

The templates for airplanes, guitars, guns, knives, and cars are generated automatically using segmentations of Yi et al. (2016). For mugs, we start with an automatic template and add a hole in the handle and a void in the interior. To demonstrate a template with distinct parts, for cars, we use the segmentation during training, computing Chamfer and normal losses for wheels and body separately. For bottle and bathtub templates, we simply place two and and five cuboids, respectively.

With a generic sphere template, we produce a compact piecewise-smooth surface of comparable quality to the more conventional deformable meshes. Our algorithmic construction of category-specific templates, however, enables higher-quality reconstruction of sharp features and details.

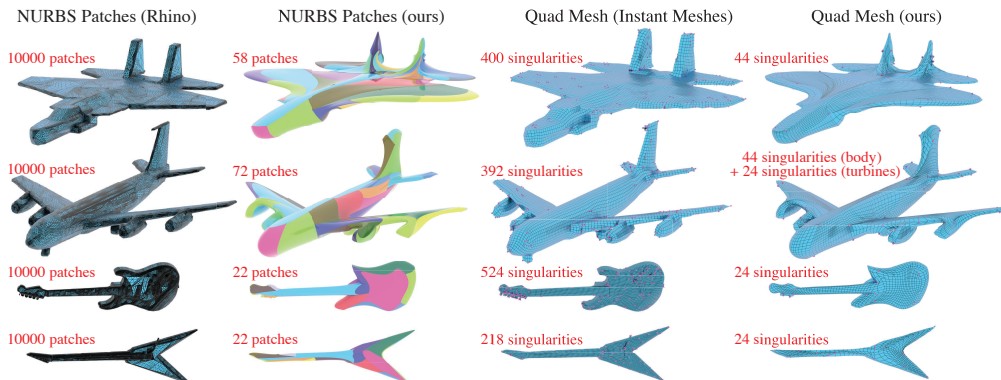

NURBS Patches (Rhino)    NURBS Patches (ours)    Quad Mesh (Instant Meshes)    Quad Mesh (ours)

Figure 6: We convert two airplane and two guitar models to NURBS patches using Rhino 3D and compare to our Coons patches. Corresponding Coons patches across models of the same category are the same color. We also show quad meshes generated using Instant Meshes (Jakob et al., 2015) and those from our patch decompositions. Singular points are in pink. Our representation is much more compact and hence easily editable. We produce fewer singularities and only in known places. Furthermore, unlike with other methods, our patches and quad meshes are consistent across models.

We demonstrate our method's ability to incorporate details from different views. We show our output when given a single view of an airplane as well as when given an additional view. The combined model incorporates elements not visible in the original view.

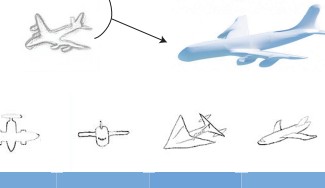

We also test on real sketches. Each artist was asked to sketch an airplane without having seen any sketches from our dataset. The results are similar to those on synthetic sketches, demonstrating that our dataset is reflective of choices that humans make when sketching 3D objects.

### 4.3 3D MODEL INTERPOLATION

Our representation is naturally well-suited for interpolating between 3D models. As each model is composed of a small number of consistently placed patches, we linearly interpolate the patch parameters (e.g., vertex positions) to generate models "between" our network outputs. We also perform interpolation in the latent space learned by our model (the output of the first 1024-dimensional hidden layer). While latent-space interpolation is similar to patch-space, each interpolant better resembles a realistic model due to the learned priors. We demonstrate patch-space (right) and latent-space (left) interpolation between two car models.

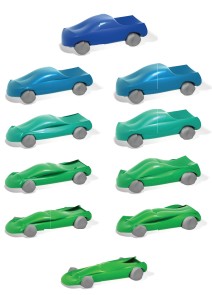

### 4.4 QUAD MESHING AND NURBS DECOMPOSITION

Many algorithms have been designed for converting triangle meshes into NURBS patches (Krishnamurthy & Levoy, 1996; Eck & Hoppe, 1996; Bernardini et al.). By fitting a template to a 3D model, our method automatically converts the model to a set of Coons patches. Moreover, our Coons patches are placed *consistently*, thus establishing correspondences between models. In Fig. 6, we compare our Coons patches to NURBS patches automatically obtained using Rhino 3D, a commercial CAD program. Our decomposition is significantly more sparse and usable for further editing.

A common task in computer graphics is converting surfaces into quad meshes. We can easily obtain a quad mesh from our patch decomposition by simply subdividing (see §A.4 for details). We compare our quad meshes to those produced by Instant Meshes (Jakob et al., 2015), a recent quad meshing algorithm, in Fig. 4.4. It is commonly desired to minimize the number of singular points (vertices with valence not equal to four) and to consistently position them in a quad mesh. Our algorithm achieves both of these goals—the number and location of singularities is determined by our template. While the Instant Meshes results contain between 215 and 524 singular points, our quad meshes contain only 24 (guitars), 44 (airplane without turbines), or 68 (airplane with turbines).

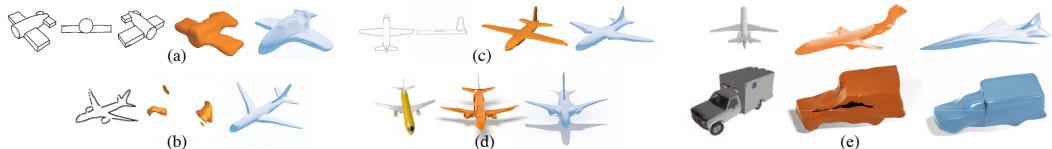

Figure 7: Comparisons to (Delanoy et al., 2018) using their data (a) and our data (b), to (Lun et al., 2017) (c), to AtlasNet (d), and to Pixel2Mesh (e). Our results are in blue for each comparison.

## 4.5 COMPARISONS

In Fig. 7, we compare to the sketch-based reconstruction methods of Delanoy et al. (2018) (a) and Lun et al. (2017) (c). Although we use the same species of input used to train these methods rather than attempting to re-train their models on our data, the visual quality of our predictions is comparable to theirs. Moreover, our output representation sparsely captures smooth and sharp features, independent of resolution. In contrast, Delanoy et al. (2018) produce a $64^3$ voxel grid—a dense, fixed-resolution representation, which cannot be edited directly and offers no topological guarantees. In Fig. 7b, we evaluate their system on contours from our dataset, demonstrating that our task of reconstruction with a prior on class (airplane) rather than structure (cylinders and cuboids) is misaligned with theirs: Since our data is not well-approximated by CSG models, their method cannot extract meaningful output.

Although Lun et al. (2017) ultimately produce a mesh, they perform a computationally expensive post-processing procedure, since their forward pass returns a labeled point cloud. Our method directly outputs the parameters for surface patches with no further optimization. Additionally, their final mesh contains more components (triangles) than our output (patches), making it less useful for editing.

In Fig. 7d, we compare to AtlasNet (Groueix et al., 2018). We retrain our model with their renderings, using the 54-face sphere template. While our 3D reconstructions capture the same degree of detail, they do not suffer from topological defects. In particular, AtlasNet's surfaces contains many patch intersections and holes. Extracting a watertight mesh would require significant post-processing. Additionally, each patch in our representation is parameterized sparsely by control points, in contrast to AtlasNet's patches, which must be sampled using a deep decoder network.

In Pixel2Mesh, Wang et al. (2018b) input an image and output a triangle mesh. We train our sphere template models on their data. While their meshes have 2466 vertices (*7398 degrees of freedom*) we output 54 patches (*816 degrees of freedom*)—a more editable and interpretable representation. As shown in Fig. 7e, the low dimensionality of our output is not at the expense of expressiveness.

## 5 DISCUSSION AND CONCLUSION

As 3D data become more readily available, the need for processing, modifying, and generating 3D models in a usable fashion increases. While the quality of results produced by deep learning systems improves, it is necessary to think carefully about their format, particularly with respect to existing use cases. By carefully designing representations together with compatible learning algorithms, we can harness these data for the purpose of simplifying workflows in design, modeling, and manufacturing.

Our system is a step toward practical 3D modeling assisted by deep learning. Our sparse patch-based representation is close to what is used in artistic and engineering practice, and we accompany it with new geometric regularizers that greatly improve the reconstruction process. Unlike meshes or voxel occupancy functions, this representation can easily be edited and tuned after 3D reconstruction, and it captures a trade-off between smoothness and sharp edges reasonable for man-made shapes.

Our work suggests several avenues for future research. While we use pre-trained networks to generate sketch data, we could couple the training of these pieces to alleviate dependence on sketch–3D model pairs. We also could leverage literature in computer-aided geometric design to identify other structures amenable to learning. Particularly, multiresolution representations (e.g., subdivision surfaces) might enable learning both high-level smooth structure and geometric details like filigree independently.

Perhaps the most important challenge remaining from our work—and others—involves inferring shape *topology*. While our system supports structural variability and modular parts, scaling this towards completely learned topology is nontrivial. This limitation is reasonable for shape classes we consider, but generic sketch reconstruction requires automatically adding and connecting patches.

## 6 ACKNOWLEDGEMENTS

The MIT Geometric Data Processing group acknowledges the generous support of Army Research Office grant W911NF2010168, of Air Force Office of Scientific Research award FA9550-19-1-031, of National Science Foundation grant IIS-1838071, from the CSAIL Systems that Learn program, from the MIT–IBM Watson AI Laboratory, from the Toyota–CSAIL Joint Research Center, from a gift from Adobe Systems, from an MIT.nano Immersion Lab/NCSOFT Gaming Program seed grant, and from the Skoltech–MIT Next Generation Program. This work was also supported by the National Science Foundation Graduate Research Fellowship under Grant No. 1122374. We acknowledge the support of the Natural Sciences and Engineering Research Council of Canada (NSERC) grant RGPIN-2019-05097 ("Creating Virtual Shapes via Intuitive Input") and from the Fonds de recherche du Québec - Nature et technologies (FRQNT) grant 2020-NC-270087.

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

## A APPENDIX

### A.1 CHANGE OF VARIABLES FOR VARIATIONAL CHAMFER DISTANCE

We derive the expression for area-weighted variational Chamfer distance:

$$\text{Ch}_{\text{dir}}^{\text{var}}(P,M) = \frac{1}{\text{Area}(P)} \int_P \inf_{y \in M} d(x,y) dx \tag{12}$$

$$= \frac{1}{\text{Area}(P)} \int_0^1 \int_0^1 \inf_{y \in M} d(P(s,t),y) |J(s,t)| ds dt \tag{13}$$

$$= \frac{1}{\text{Area}(P)} \cdot \frac{1}{\text{Area}(\square)} \int_0^1 \int_0^1 \inf_{y \in M} d(P(s,t),y) |J(s,t)| ds dt \tag{14}$$

$$= \frac{1}{\text{Area}(P)} \mathbb{E}_{(s,t) \sim \mathcal{U}_\square} \left[ \inf_{y \in M} d(P(s,t),y) |J(s,t)| \right] \tag{15}$$

$$= \frac{\mathbb{E}_{(s,t) \sim \mathcal{U}_\square} [\inf_{y \in M} d(P(s,t),y) |J(s,t)|]}{\mathbb{E}_{(s,t) \sim \mathcal{U}_\square} [|J(s,t)|]} \tag{16}$$

### A.2 ABLATION STUDY

We perform an ablation study of our method on an airplane model, demonstrating the effect of training without each term in our loss function as well as the difference between a category-specific template, a 54-patch sphere template, and a lower resolution 24-patch template. The results are shown in Fig. 8. We also show an ablation results on the knife model in Fig. 9.

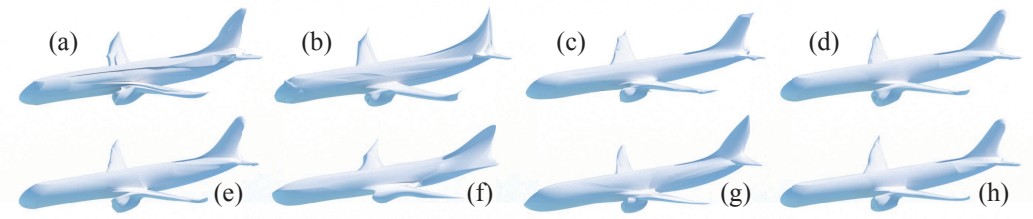

Figure 8: An ablation study of our model without normal alignment loss (a), without collision detection loss (b), without patch flatness loss (c), without template normal loss (d), without symmetry loss (e), as well as using 24-patch (f) and 54-patch (g) sphere templates compared to the final result (h).

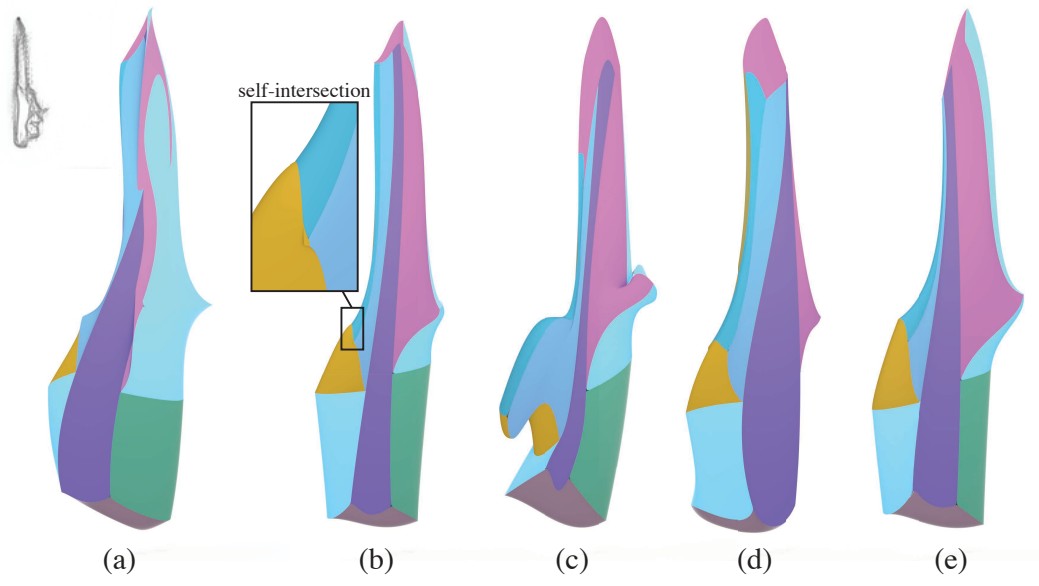

Figure 9: An ablation study of our model without normal alignment loss (a), without collision detection loss (b), without patch flatness loss (c), without template normal loss (d), compared to the final result (e).

The ablation study demonstrates the contribution of each component of our system method to the final result. Training without collision detection loss results in patch intersections. Omitting the normal loss causes the 3D surface to suffer in smoothness. Patch flatness and and template normal losses encourage patch seams to align to sharp features. While both sphere templates capture the geometry, using more patches captures greater details, and using a non-generic template further improves the model.

We additionally provide a quantitative ablation study for the knives and airplanes category in Table 1. We train a model without each of the normals, flatness, template normals, and collision loss terms for 50000 steps. for each model, we report the average Chamfer loss, normals loss, flatness loss, and template normals loss (as defined in §3.2) on the test set, in addition to the average number of intersecting patch pairs. The results demonstrate that each loss term contributes to our full model without sacrificing reconstruction quality.

| Model | Ch. Loss | Norm. Loss | Flat. Loss | Temp. Norm. Loss | # Int. Pairs |
|---|---|---|---|---|---|
| airplanes (full) | 0.0118 | 0.725 | 0.0000732 | 0.792 | 0.0220 |
| airplanes (no norm.) | 0.0105 | 1.09 | 0.0000941 | 0.935 | 0.0357 |
| airplanes (no flat.) | 0.0119 | 0.748 | 0.000254 | 0.866 | 0.0302 |
| airplanes (no temp. norm.) | 0.0114 | 0.723 | 0.0000734 | 0.889 | 0.0110 |
| airplanes (no coll.) | 0.0112 | 0.727 | 0.000113 | 1.07 | 3.09 |
| knives (full) | 0.0124 | 0.635 | 0.0000950 | 0.669 | 0 |
| knives (no norm.) | 0.0124 | 1.12 | 0.0000902 | 1.46 | 3.95 |
| knives (no flat.) | 0.0122 | 0.648 | 0.00132 | 0.733 | 0.0263 |
| knives (no temp. norm.) | 0.0126 | 0.671 | 0.0000886 | 1.03 | 0.132 |
| knives (no coll.) | 0.0122 | 0.631 | 0.0000963 | 0.722 | 0.0263 |

Table 1: Quantitative ablation study of our loss function.

## A.3 QUANTITATIVE COMPARISON TO PIXEL2MESH

We compare to Pixel2Mesh quantitatively in Table 2. We select 2500 random test set views and compute Chamfer distance using 5000 sampled points. While obtain comparable Chamfer distance values, our representation is significantly more compact, editable, and less prone to non-manifold artifacts.

| Category | CD | | DOF | |
|---|---|---|---|---|
| | P2M | ours | P2M | ours |
| airplane | 0.022 | 0.025 | 7398 | 816 |
| car | 0.018 | 0.022 | 7398 | 816 |

Table 2: Quantitative comparison to Pixel2Mesh. We report Chamfer distance (CD) and degrees of freedom in the representation (DOF). We obtain comparable Chamfer distance using a representation that is an order of magnitude more compact and without non-manifold artifacts.

## A.4 QUAD MESH GENERATION

Given a collection of Coons patches, we obtain a quad mesh by subdividing each patch into some number of quads. In order for the quad mesh to be uniform, we determine the number of subdivisions for each boundary curve of each patch by solving an integer linear problem (ILP). Our free variables are the number of subdivisions per curve, and the objective function encourages the number of subdivisions to be proportional to the arc length of each curve. The constraints ensure that opposite curves for each patch are subdivided into an equal number of segments, and when one curve contains multiple other curves (due to edge junctions), the numbers of subdivisions are compatible. In particular, we solve the following ILP:

$$\text{minimize} \quad \sum_{i \in I} (x_i - \alpha s_i)^2$$
$$\text{s.t.} \quad x_i = x_{\mathcal{O}(i)} \qquad\qquad \forall i \in \mathcal{E}$$
$$x_i = \sum_{j \in \mathcal{J}(i)} x_j \qquad\qquad \forall i \in \mathcal{E}$$
$$x_i \in \mathbb{Z}^+ \qquad\qquad \forall i \in \mathcal{E},$$

where $x_i$ is the number of subdivisions for curve $i$, $s_i$ is the arc length of curve $i$, $\mathcal{E}$ contains all of the curves, $\mathcal{O}$ maps a curve to the opposite curve in its patch, $\mathcal{J}$ maps a curve to a list of curves contained within it, and $\alpha$ is a parameter that controls the coarseness of the resulting quad mesh.

We solve the ILP using MOSEK, subdivide our Coons patches to obtain a quad mesh, and then perform surface-preserving Laplacian smoothing in MeshLab for 10 iterations with maximum angle displacement of $5°$.

### A.5 ADDITIONAL QUALITATIVE RESULTS

We show models from the test set for each shape category produced by our full model trained with category-specific templates.

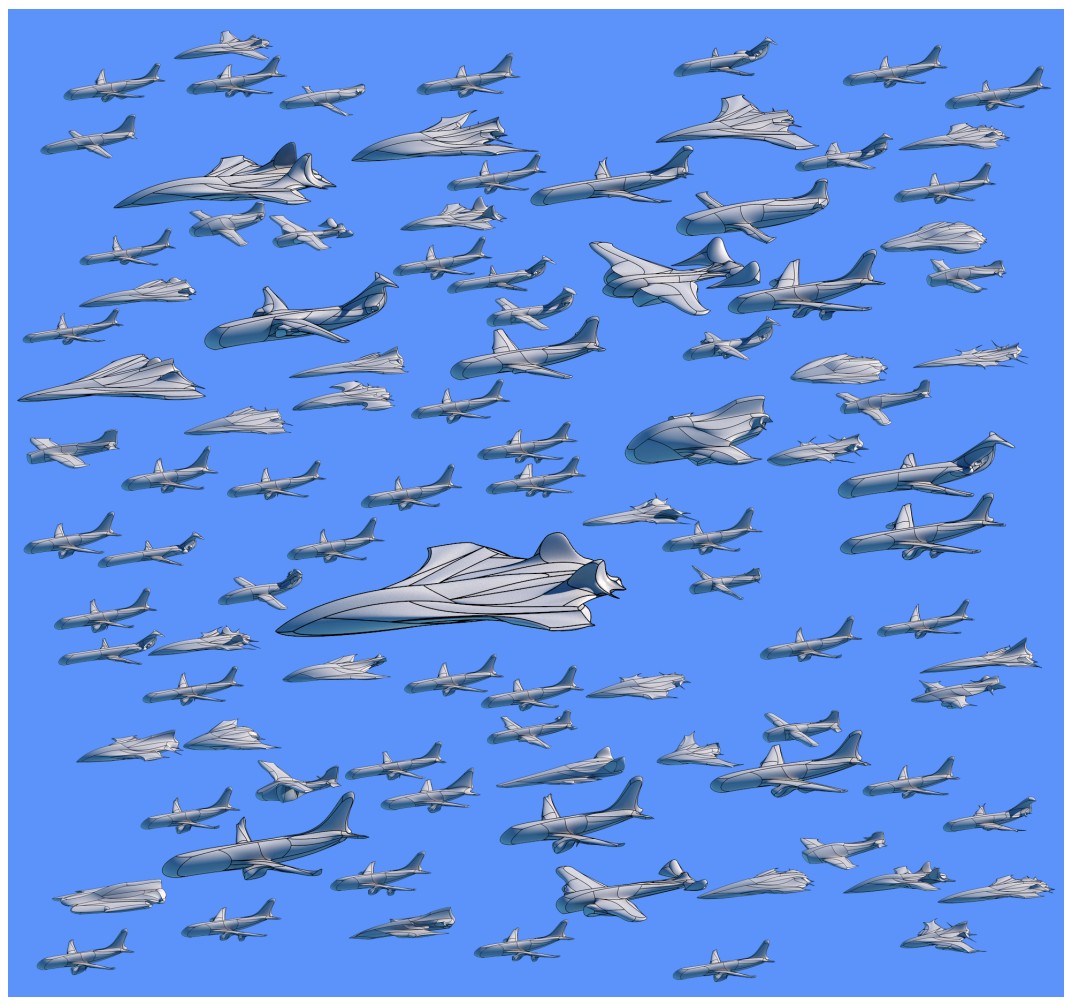

Figure 10: Airplanes from the test set.

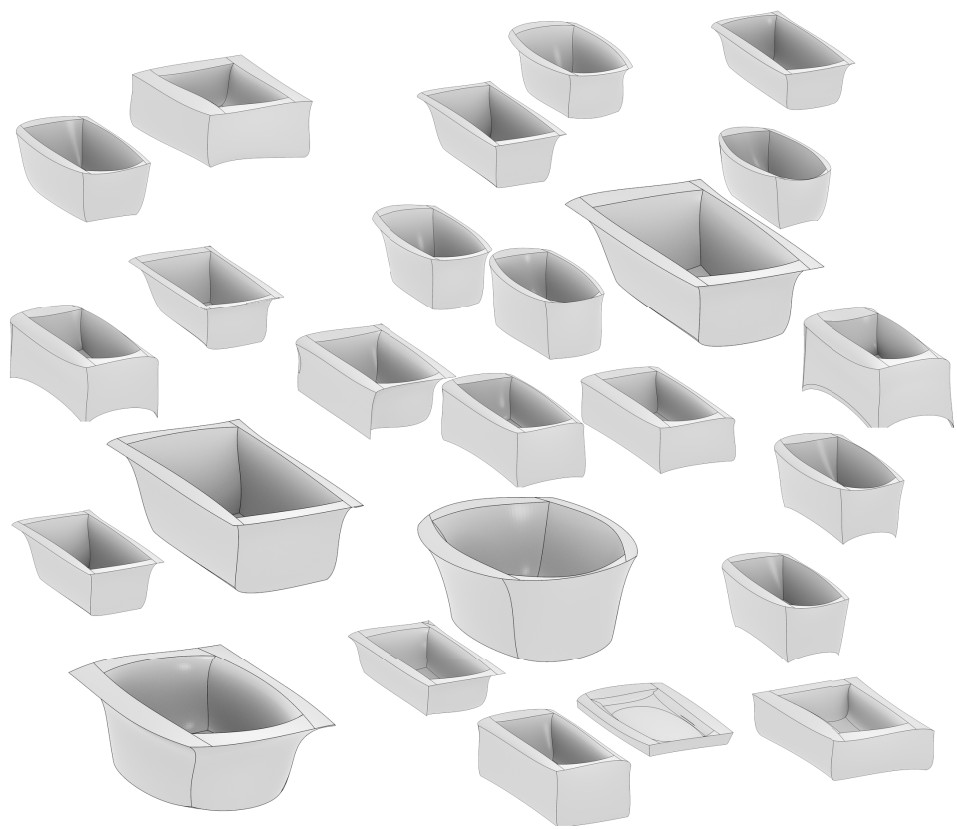

Figure 11: Bathtubs from the test set.

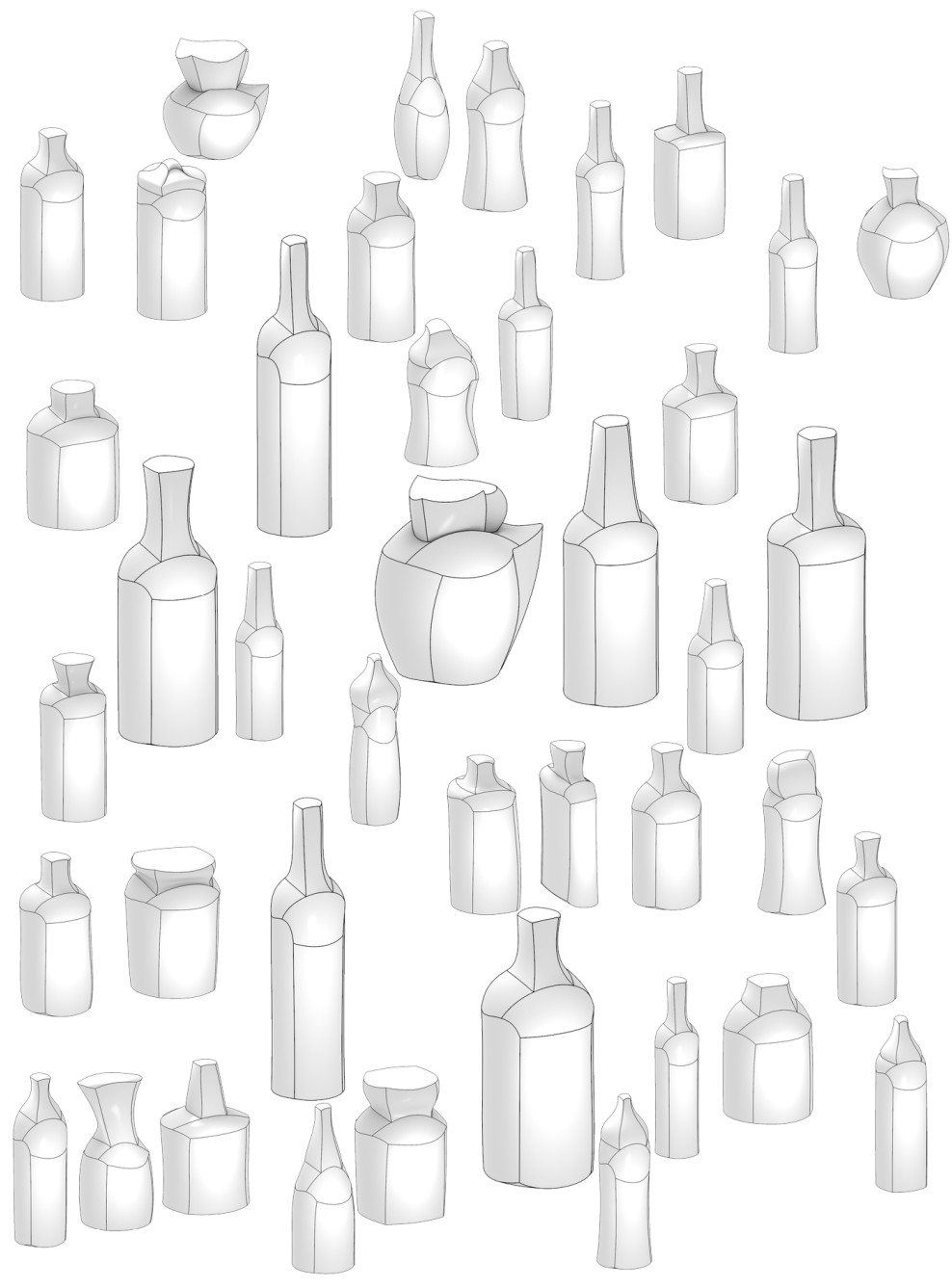

Figure 12: Bottles from the test set.

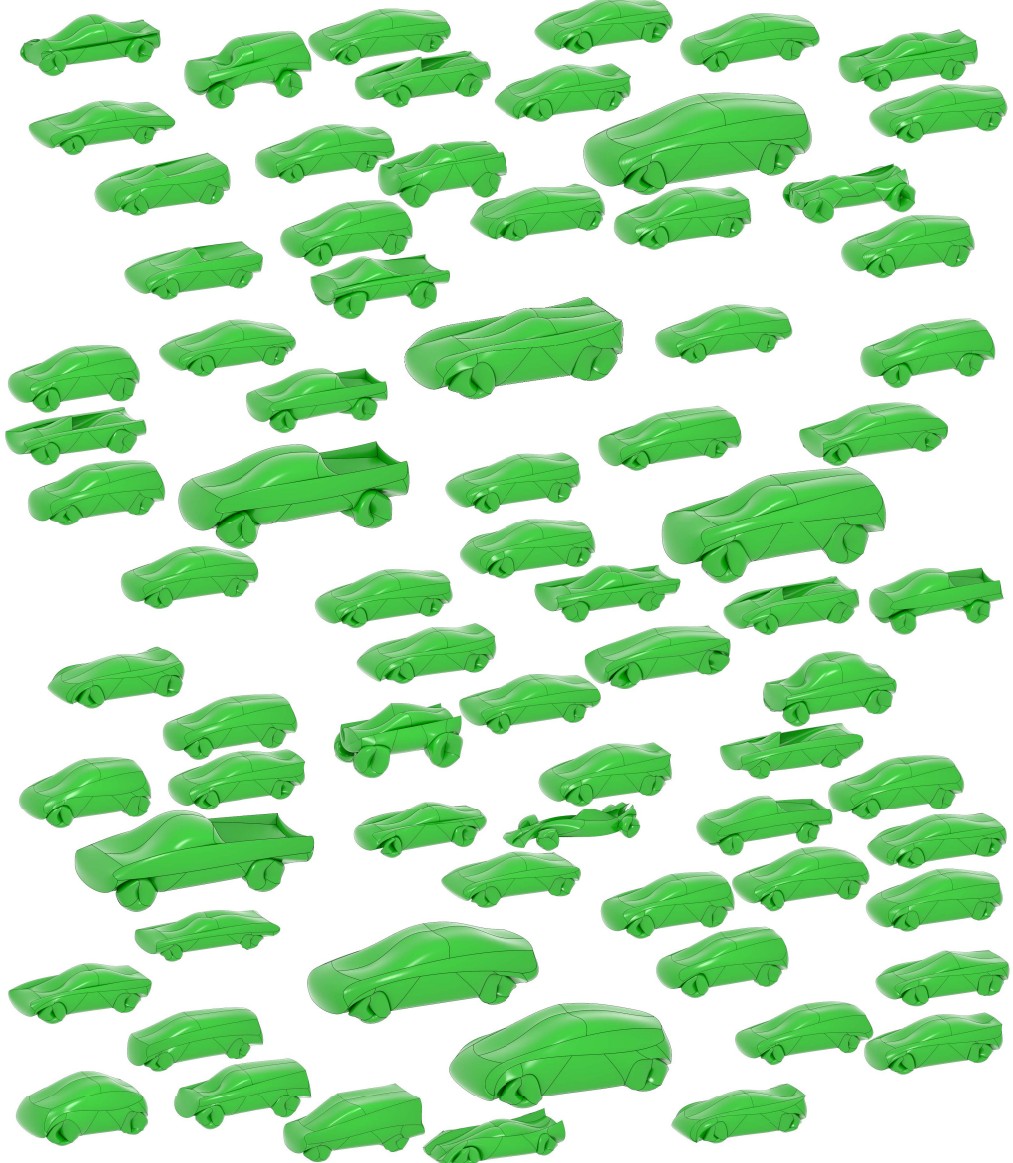

Figure 13: Cars from the test set.

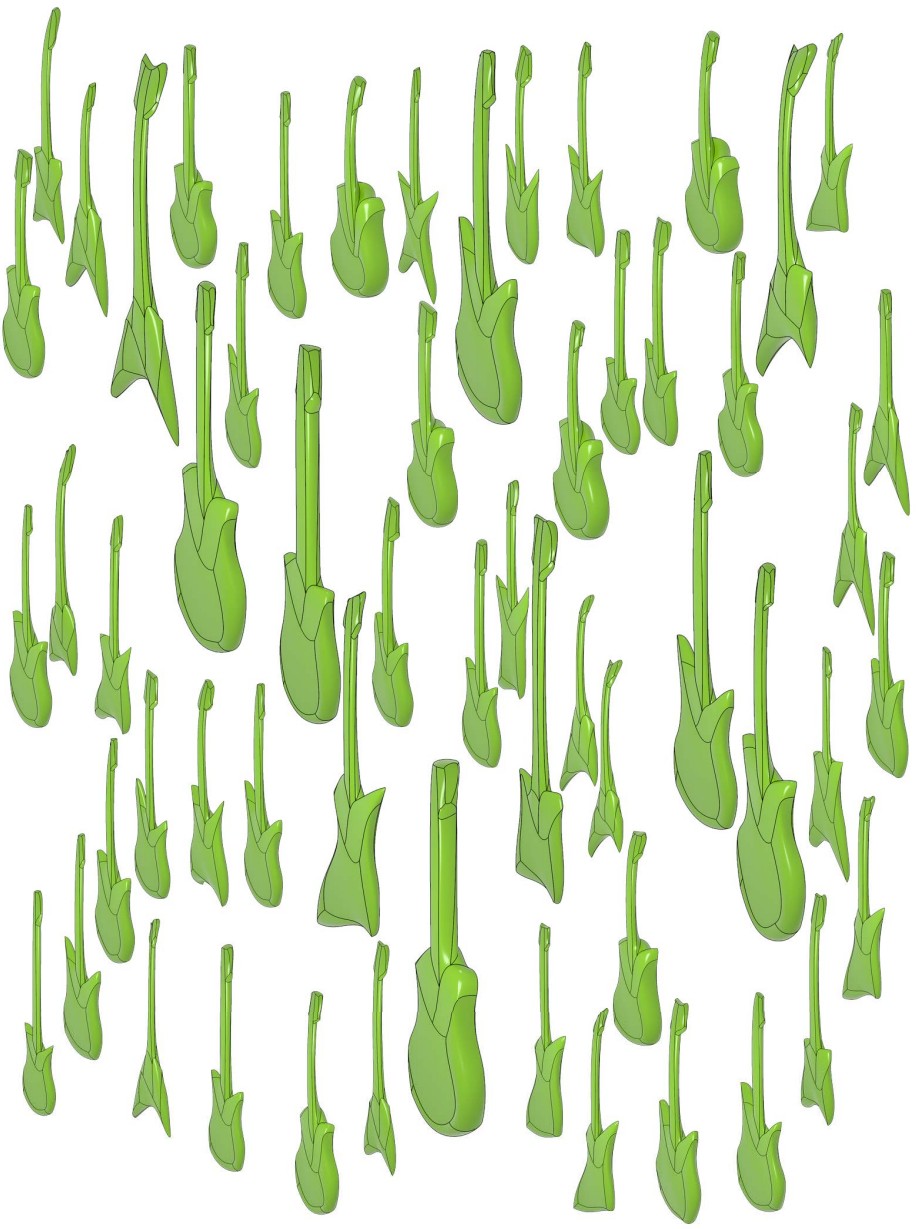

Figure 14: Guitars from the test set.

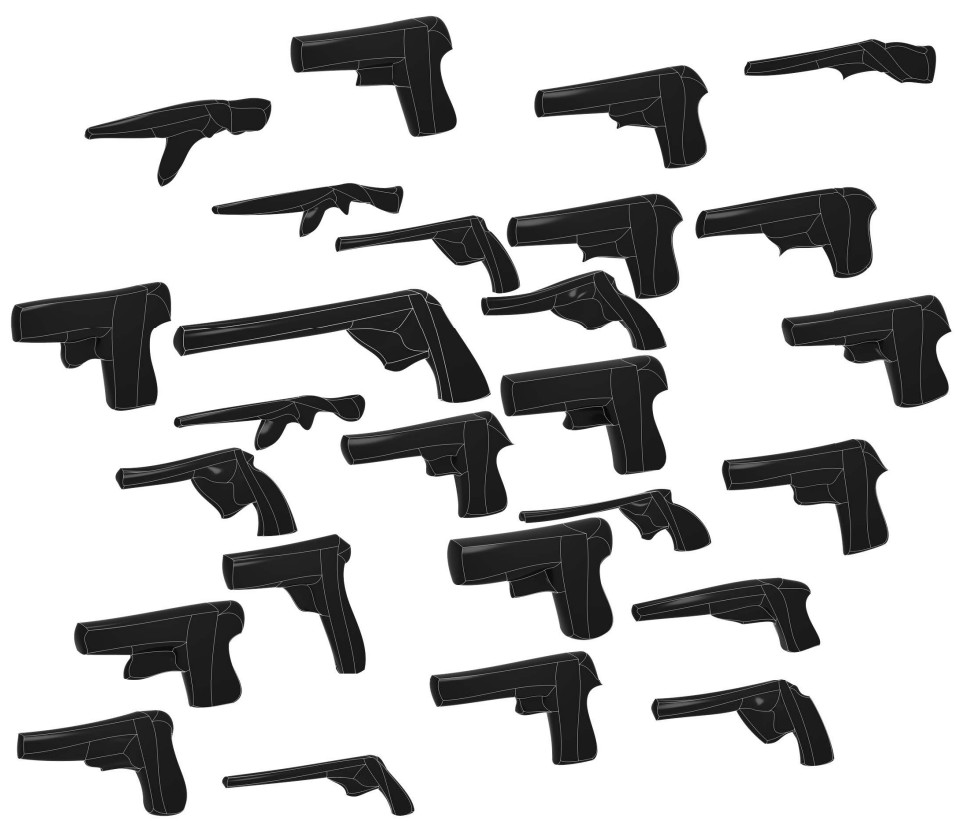

Figure 15: Guns from the test set.

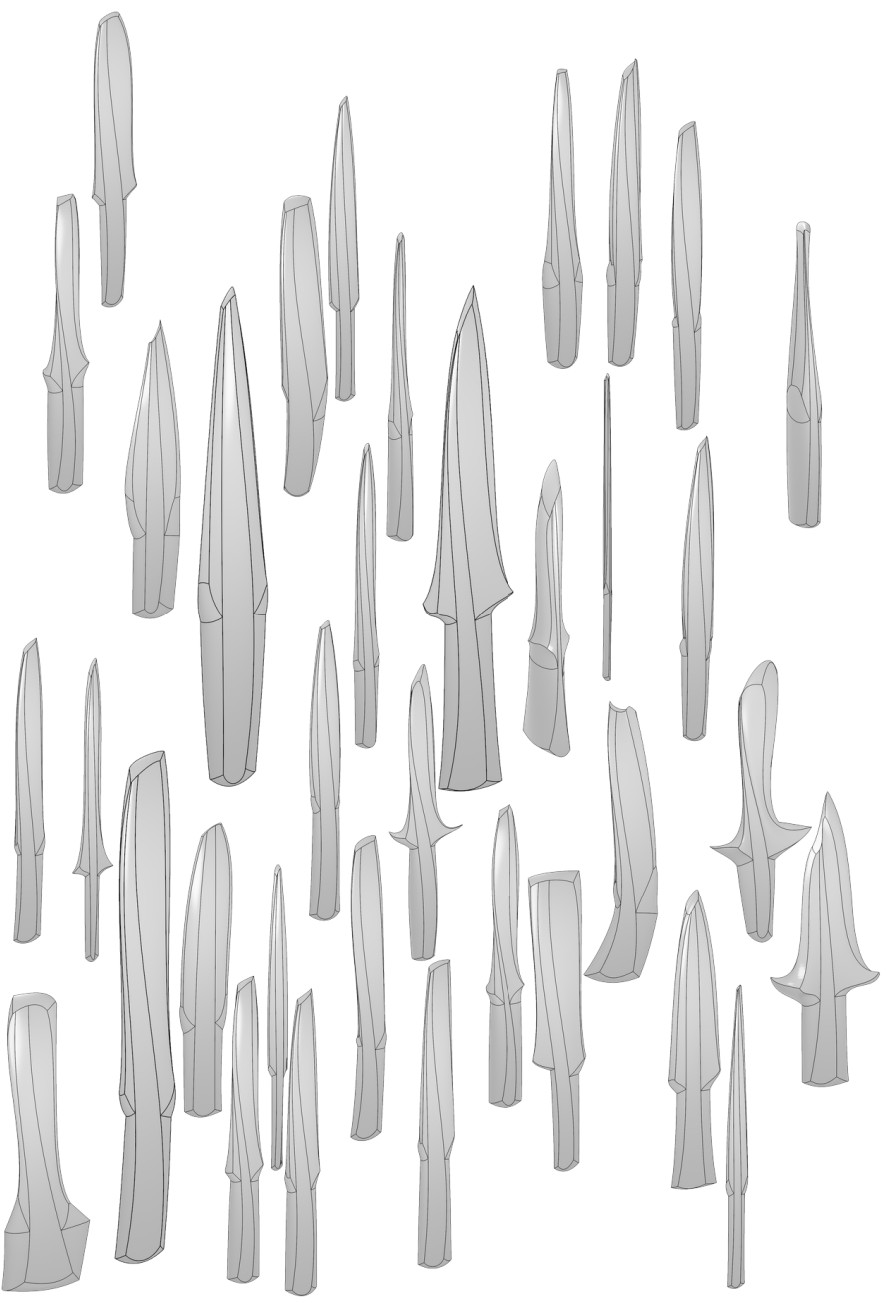

Figure 16: Knives from the test set.

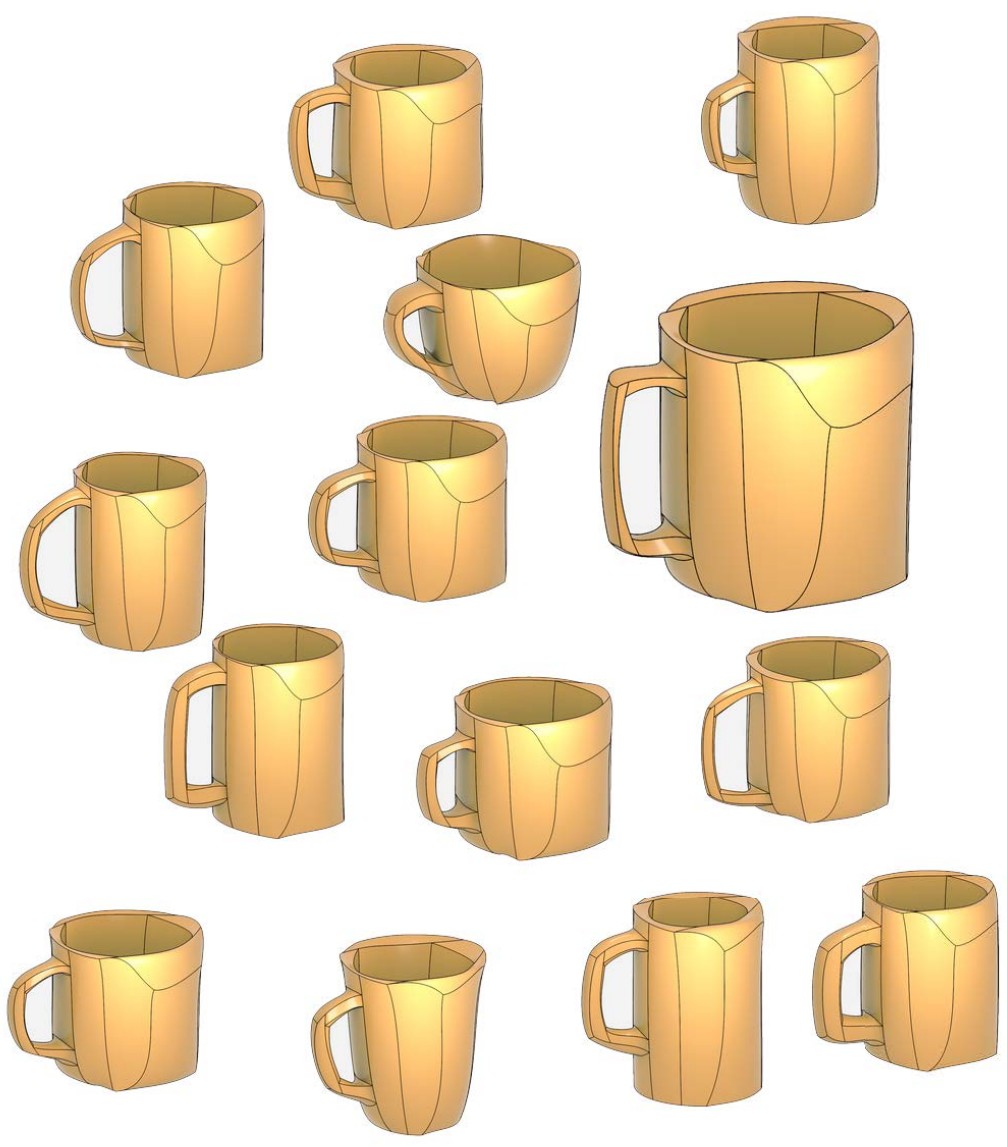

Figure 17: Mugs from the test set.

