# OpenReview forum: "Learning Manifold Patch-Based Representations of Man-Made Shapes"
_ICLR.cc/2021/Conference — ICLR 2021 Poster_

### Official Review · AnonReviewer3 · 2020-10-25
**Claims need to be backed up.**

**Rating:** 7
**Confidence:** 5

**Review:**

I like the idea of this paper and think that finding new representations of 3D shapes in deep learning approaches is important and interesting. However I have a number of concern with respect the claims of this paper, and the lack of sufficient evidence backing them up:

-> What does "sampling uniformly from patches is difficult" mean? Can you clarify this statement both here, and in the paper?

-> You provide all these detail about how you make the chamfer distance and normal alignment work for patches, but then say you can triangulate the patches anyway. If you can triangulate the patches why can you not directly apply the chamfer distances and normal alignment to points uniformly sampled over these triangles?  See: "GEOMetrics: Exploiting Geometric Structure for Graph-Encoded Objects" or "Point2Mesh: A Self-Prior for Deformable Meshes" for detail on how this is done. Would this not be easier? This would probably also be true for the patch flatness regularizer, where instead you could condition adjacent faces to have low curvature such as in the smoothness loss of "Neural 3D Mesh Renderer". This is not to say that the approaches you have taken are wrong, but you are proposing alternatives to solutions that already exist, perform well, and are naturally applicable to your setting. I think is it very important that you compare your approaches to these in some way in the paper. An experiment to this effect would go a long way to increasing my score.


-> The ablation you provide in the supplemental is not sufficient. Demonstrating the difference over a single model does not provide me a good understanding of the differences. You should provide a visual ablation over a larger number of  models, and in addition provide a quantitative ablation of your metrics( chamfer distance, normal alignment) over your whole test set. This is absolutely necessary. Again showing this here and in the paper will go a long way to increasing my score.

-> You provide an explicit definition of patches, but no intuition as to what they are. A reader can go and look up coons patches independently, but it would be helpful to provide some help here. A diagram might be helpful to explain this. Patches seem to be quite similar to charts as referenced in "AtlasNet: A Papier-Mâché Approach to Learning 3D Surface Generation
" and "3D Shape Reconstruction from Vision and Touch". It might be helpful to disambiguate this in the related work. I realize you touch lightly on this in the results section, though this it too late in the paper to establish the understanding on your approach.

-> There is a lack of quantitative comparison in this paper. I see that you compare visually to Pixel2Mesh and AtlasNet, but this is only over a small number of models. A reader has no way to determine if these results have been hand picked to make your approach look superior. You need to provide a comparison to contemporary methods for 3D reconstruction from images. Here Pixel2Mesh and AtlasNet are probably not sufficient either as better performing methods have since been released. The comparison in the supplemental is not sufficient, It should be over a large number of categories, and methods.

-> You write: "Unlike meshes or voxel occupancy functions, this representation can easily be edited and tuned after 3D reconstruction". Meshes are easily edited in 3D design software and also can be converted to NURBS. In addition voxels and occupancy functions can be converted into mesh representations and so also editable in these software as well. In fact I believe 3D deep learning frameworks exists which handle this conversion natively. I think you should be clear about these claims and make them less strong.

I want to be convinced that the proposed approach works better that what is already available , or at least better in some specific domain. The paper in its current form does not do this due to the concerns raised above. I am willing to revise my score if they are properly addressed.

---------------------------------------------------------------------------------------------------------------------------

I believe that the authors have adequately addressed my concerns, and have made a good effort to improve the quality of their work and so I am raising my score.

---

> ### Author Response · Authors · 2020-11-11
> **Response**
>
> We thank the reviewer for their comments and suggestions. We are committed to improving our paper and can easily address all concerns within the rebuttal period. Below, we address each comment and indicate in **bold** additional experiments that we will post within several days. We hope that these will allay any concerns about our work and convince the reviewer that it will be a welcome contribution to the ICLR community. **If there are additional experiments that we can perform to further support our case, please let us know.**
>
> > What does "sampling uniformly from patches is difficult" mean?
>
> Sampling uniformly in the parameter space of a patch (the unit square) does not yield uniform samples on the surface (with respect to the area measure of the surface as a geometric object)---regions with higher curvature end up oversampled. A closed form arc-length parameterization does not even exist for a Bezier curve, and so a uniform parameterization for even a one-dimensional patch is difficult to compute. Many works try to address this issue by designing custom sampling procedures [1, 2, 3]. These schemes are typically computationally expensive and/or incompatible with a differentiable learning pipeline.
>
> > If you can triangulate the patches why can you not directly apply the chamfer distances and normal alignment to points uniformly sampled over these triangles?
>
> Triangulating the patches would result in the same non-uniform sampling. While we would be able to sample uniformly from each triangle, the triangles themselves would not be uniformly distributed on the surface. Thus, the sampling issue would simply be deferred to when we sample from the triangles, not circumvented.
>
> > You should provide a visual ablation over a larger number of models, and in addition provide a quantitative ablation of your metrics( chamfer distance, normal alignment) over your whole test set.
>
> **We will add both a quantitative ablation as well as more qualitative examples.**
>
> > You provide an explicit definition of patches, but no intuition as to what they are.
>
> **We will modify Fig. 3 to illustrate how control points define a Coons patch.** Unlike charts in AtlasNet, which are parameterized using neural networks, our patches are explicitly defined via a small number of control points. Additionally, while AtlasNet charts are free to overlap each other as long as they cover the surface, our patches are equipped with a topology, which ensures that adjacent patches meet along their boundary curves.
>
> > You need to provide a comparison to contemporary methods for 3D reconstruction from images.
>
> We compare to Pixel2Mesh to demonstrate that our method is able to achieve similar reconstruction quality to a mesh-based method while using a significantly smaller number of elements (our number of patches is two orders of magnitude smaller than their number of faces), yielding a representation that is significantly more useful for modeling tools. We compare to AtlasNet as it is the only patch-based deep 3D reconstruction work that we are aware of. While newer reconstruction methods achieve better reconstruction results, we were unable to identify any papers that reconstruct shapes using a sparse set of editable elements. We respectfully ask the reviewer for specific pointers to methods that propose a similar representation.
>
> > Meshes are easily edited in 3D design software and also can be converted to NURBS.
>
> Representing surfaces using sparse collections of parametrically-defined patches has been a long open problem in the modeling and design literature (called “the holy grail of NURBS design” [4]). While there have been many attempts to design procedural algorithms for converting triangle meshes to NURBS patches automatically, these suffer from many artifacts. **We will add experiments that compare our patches with those obtained automatically from a source mesh.**
>
> [1] Wang, Jiang, et al. “Intelligent sampling for the measurement of structured surfaces.” Measurement Science and Technology 2012.
>
> [2] Elkott, Elmaraghy, and Elmaraghy. “Automatic sampling for CMM inspection planning of free-form surfaces.” International Journal of Production Research 2002.
>
> [3] Hernandez-Mederos and Estrada-Sarlabous. “Sampling points on regular parametric curves with control of their distribution.” Computer Aided Geometric Design 2003.
>
> [4] https://www.grasshopper3d.com/forum/topics/auto-nurbs-surfacing-of-meshes

---

> > ### Comment · AnonReviewer3 · 2020-11-11
> > **Reply**
> >
> > Thank you for addressing the comments. From what you have written, if you follow through with your proposed changes I do not see any reason not to raise your score. One sticking point I have though, and this may still be a misunderstanding on my part, is the triangulation of the patches. While the triangulation may not be uniform this does not matter as you can sample points from the triangles with respect to their relative surface area, which produces a uniform sampling of points on the surface of the shape( see the references I provided to see how this is done). Maybe I am missing something here though. To help clear this up could you provide an example of one of your predicted shapes, both before and after triangulation so I can see what you mean by "the triangles themselves would not be uniformly distributed on the surface"? Also is this triangulation differentiable?

---

> > > ### Author Response · Authors · 2020-11-12
> > > **Clarification**
> > >
> > > Thank you for the response!
> > >
> > > Here is an example of a (uniform) triangulation of the parameter domain and the corresponding triangulated Coons patch, as computed during our training procedure:
> > >
> > > https://i.ibb.co/YRJtkJg/triangulation.png
> > >
> > > Even though the triangles in the parameter domain are identical, in the 3D patch the area of each triangle is different due to the nonuniform sampling issue we describe. You are correct that we can sample uniformly from this mesh according to the surface area of each triangle. However, the discretization step would unnecessarily complicate and introduce error into our computations, relative to our equally (if not more) efficient and accurate alternative---each triangle does not lie perfectly on the smooth surface, and we would need to calculate/interpolate normals. Contrastingly, our explicit parametric representation admits analytical computation of the relevant quantities in closed form without discretization or approximation.
> > >
> > > Our collision detection step, which uses the triangulation, is indeed differentiable. We take a fixed triangulation of the parameter space (a unit square) and simply compute the image of each vertex under the Coons patch map, keeping the original connectivity.
> > >
> > > Does this address your concerns about triangulation? If not, it would be our pleasure to provide additional clarifications, illustrations, or experiments.

---

> > > > ### Comment · AnonReviewer3 · 2020-11-12
> > > > **Reply**
> > > >
> > > > Okay, yes I understand, thanks you. I would strongly suggest you make statements to this effect in the paper when introducing your losses, and cite the relevant references. If you wanted to go the extra mile, quantitatively validating these statements would be helpful, however I understand this may not be feasible in the review timeline.

---

> > > > > ### Author Response · Authors · 2020-11-23
> > > > > **Follow-up**
> > > > >
> > > > > Thank you again for your comments and suggestions. We have uploaded a revised draft that includes all of our promised changes.
> > > > >
> > > > > If we have successfully addressed your questions, we would strongly appreciate an increased score. Otherwise, please let us know what experiments and/or revisions we can provide to allay your concerns.

---

### Official Review · AnonReviewer1 · 2020-10-26
**A nice study, with flaws in quantitative analysis**

**Rating:** 7
**Confidence:** 3

**Review:**

The paper proposes a self-supervised method to fit a template (represented as a union of Coons patches) to a certain 2D sketch. It derives a way to build a proper template, uses a network to predict the patches' parameters, and proposes a union of different losses. The qualitative results of the method are shown in several different objects.

PROS
- Losses: I appreciated the effort of the authors in formulating an ensemble of losses for this specific representation. I think they will be useful for future works with similar representations.
- Versatility: this representation permits working with a large variety of different classes of objects (also disconnected ones, but not topological changes), at an arbitrary resolution and predicts a small set of values.

The paper does not clearly state if the code and the data will be made publicly available, but since the code is attached to the submission I assume this is in the will of the authors, and this is also a contribution.

Finally, the paper is well organized and I like the presentations.

CONS
- Quantitative analysis: the only quantitative evaluation is reported in the supplementary material. I think this is the major weakness of the paper. I would suggest having more experiments to analyze the performance of the method. In particular, the ablation is a critical part since many losses are involved; I would suggest providing some quantitative measures, for example showing how each loss improves a particular aspect (e.g. the distance from the ground truth, the difference between normals, the intersection area, ...).

I would also suggest highlighting the output patches on some qualitative results.

Minor fixes:
- Equation 4, fix the pedix of $\mathcal{U}_{\[0,1\]^2}$
- Equation 13 (Sup.Mat), fix the Integral limits locations
- I think "P2P-NET: Bidirectional Point Displacement Net for Shape Transform"(Yin et al., 2018) worth a mention since solves (among the others) a similar problem


PRE-REBUTTAL RATING
I am favorable to accept the paper because it provides several tools that will be useful for future works, and I think it has interest points for the Computer Graphics community. For the acceptance I would recommend adding some more quantitative analysis, both for ablation and for the shown results.
----------
FINAL-RATING

I read the other reviews and authors' replies carefully.

First of all, I would acknowledge the effort of the authors in replying to reviewers' concerns.

About my points, I agree with other reviewers that the quantitative analysis is a bit limited (I think it is also the main criticism), but the introduced ablation and the new Figure 6 (numerically comparing against other SOTA methods) are convincing. The authors solved also my other concerns, and so I raise my score; I think the study of this representation is interesting, and well fit the audience of ICLR.

Thanks to the author for their availability, and best of luck with their work!

---

> ### Author Response · Authors · 2020-11-11
> **Response**
>
> We thank the reviewer for their comments and suggestions. Below, we address each comment and indicate in **bold** additional experiments that we will post within several days. We hope that these will allay any concerns about our work and convince the reviewer that it will be a welcome contribution to the ICLR community. **If there are additional experiments that we can perform to further support our case, please let us know.**
>
> > The paper does not clearly state if the code and the data will be made publicly available
>
> We will indeed publicly release all code and data upon publication.
>
> > In particular, the ablation is a critical part since many losses are involved
>
> **We will add a quantitative ablation study to emphasize the importance of each loss term.**
>
> > I would also suggest highlighting the output patches on some qualitative results.
>
> While we show this in Fig. 1, where we demonstrate how our representation may be edited in CAD software, we will add an additional figure.

---

> > ### Comment · AnonReviewer1 · 2020-11-13
> > **Reply**
> >
> > Thanks for your reply and to have specified that the code and data will be publicly available.
> >
> > I would clarify my point about "highlighting the output patches": I was wondering how the patches look like (i.e. where are the control points and the gluing boarders, similar to Figure 3) in the final output, like on the examples in Figure 5.
> >
> > I see that other reviewers rise points about quantitative analysis and ablations too. The analysis you proposed in the other answers would satisfy me, and in case I will be favorable to raise the score.

---

> > > ### Author Response · Authors · 2020-11-23
> > > **Follow-up**
> > >
> > > Thank you again for your comments and suggestions. We have uploaded a revised draft that includes all of our promised changes.
> > >
> > > If we have successfully addressed your questions, we would strongly appreciate an increased score. Otherwise, please let us know what experiments and/or revisions we can provide to allay your concerns.

---

### Official Review · AnonReviewer2 · 2020-10-28
**Motivation is not clear, lack of quantitative evaluation**

**Rating:** 6
**Confidence:** 4

**Review:**

This paper presents a method that leverages parametric surface patches as the fundamental representation in the task of shape modeling and reconstruction. This method requires a pre-generated template for each shape category. Several losses are specially designed to regularize the generation of the surface patches. Empirical results have demonstrated the performance of the proposed method in sketch-based shape reconstruction and 3D shape interpolation.

Pros:
- The paper shows some good results on reconstructing some simple shapes, such as bottles, mugs, bathtubs, etc.
- The idea of using parametric surface patches may reduce the parameter space that a network has to search and could potentially lead to smoother surfaces.

Cons:
- It is not clear to me why such representation is advantageous over conventional polygonal mesh-based representation.
1) Though the parametric surface patch can typically lead to smoother reconstruction, it may also suffer from the incapability of capturing fine geometric details.
2) Similar to the mesh-based approach,  the patch-based method still has to fight similar challenges, e.g. avoiding the intersections, undesirable shape distortions, etc.
3) Editing-wise, the reconstructed polygonal mesh can be easily converted to parametric surface patches to ease the following editing workflow, by registering the pre-generated template of coon patches to the reconstructed shape.

- There are no quantitative evaluations in the paper.  Also, the visual comparisons with other approaches are very limited -- only one or two examples are presented. Due to the lack of evaluations, it is difficult to determine the effectiveness of the proposed approach.

---- Final Rating ----

The revised version of the paper with additional experiments has addressed my concern on the limited advantage over prior deep 3D representation.  Fig.6 in the revised version has shown the potential of the proposed approach in generating directly usable representation in downstream applications, including CAD design and manufacturing. Hence, I would change my rating to positive.

However, the paper still lacks quantitative and qualitative evaluations. I hope it could be properly addressed in the final version.

---

> ### Author Response · Authors · 2020-11-11
> **Response**
>
> We thank the reviewer for their comments and suggestions. We are committed to improving our paper and can easily address all concerns within the rebuttal period. Below, we address each comment and indicate in **bold** additional experiments that we will post within several days. We hope that these will allay any concerns about our work and convince the reviewer that it will be a welcome contribution to the ICLR community. **If there are additional experiments that we can perform to further support our case, please let us know.**
>
> > It is not clear to me why such representation is advantageous over conventional polygonal mesh-based representation. [...] Editing-wise, the reconstructed polygonal mesh can be easily converted to parametric surface patches to ease the following editing workflow, by registering the pre-generated template of coon patches to the reconstructed shape.
>
> We respectfully disagree: Representing surfaces using sparse collections of parametrically-defined patches has been an open challenge in modeling and design for decades (called “the holy grail of NURBS design” [1]).
>
> While there have been many attempts to design algorithms for converting triangle meshes into NURBS patches [2, 3, 4], these suffer from serious artifacts, typically producing too many patches. This makes the converted models difficult to edit. **We will add experiments that compare our patches with those obtained automatically from a source mesh using the conversion functionality of modern CAD software.**
>
> > There is no quantitative evaluations in the paper. Also, the visual comparisons with other approaches are very limited -- only one or two examples are presented.
>
> We provide quantitative comparison to Pixel2Mesh in the supplementary material, Table 1. **We will also add a quantitative ablation study to emphasize the importance of our loss terms as well as a large selection of qualitative results from our test set.**
>
> If the reviewer has additional specific suggestions for quantitative evaluations, please respond on this bulletin board, and we would be happy to produce them during the discussion period.
>
> [1] https://www.grasshopper3d.com/forum/topics/auto-nurbs-surfacing-of-meshes
>
> [2] Eck and Hoppe. “Automatic Reconstruction of B-Spline Surfaces of Arbitrary Topological Type.” SIGGRAPH 1996.
>
> [3] Krishnamurthy and Levoy. “Fitting smooth surfaces to dense polygon meshes.” SIGGRAPH 1996.
>
> [4] Bernardini, Bajaj, et al. “Automatic reconstruction of 3D CAD models from digital scans.” IJCGA 1999.

---

> > ### Comment · AnonReviewer2 · 2020-11-17
> > **response**
> >
> > Thank you for the response!
> >
> > While the proposed method does have merits in generating and editing some regularly shaped objects, I'm not convinced that it has significant advantages over the state-of-the-art mesh-based or implicit function-based approaches.
> > My main concern is that the proposed method cannot produce fine-scale details and may also suffer from artifacts and self-intersection.
> > Though there is one quantitative comparison with Pixel2Mesh, the evaluations are still lacking.
> > I am expecting more comparisons with the neural implicit function-based methods both quantitatively and qualitatively.
> > Hence, I will keep my previous rating.

---

> > > ### Author Response · Authors · 2020-11-23
> > > **Clarification**
> > >
> > > We would like to clarify a subtle difference between the task addressed by our work and that solved by neural implicit methods.
> > >
> > > While deep implicit reconstruction algorithms may produce somewhat more detailed 3D output than our algorithm, our focus is on generating a 3D representation that is highly **usable** for downstream tasks in design, manufacturing, and related disciplines.
> > >
> > > To address your concern, we have added experiments (Sec. 4.4) that compare our patch decompositions to those obtained from ground truth triangle meshes using Rhino 3D, an industry-standard commercial CAD tool. Our decompositions are an order of magnitude more compact and are also consistent across models. Even assuming that a neural implicit method is able to produce a perfect reconstruction of the 3D geometry, there is **no automatic way** to convert it into a representation usable in CAD. Our algorithm is a step toward leveraging the power of deep learning for generating editable and usable shape representations.

---

### Official Review · AnonReviewer4 · 2020-11-01
**Good idea with potential novelty, but definitely needs more evaluation**

**Rating:** 4
**Confidence:** 5

**Review:**

Summary of the paper:
This paper proposes a shape generation methods where shapes are represented by a set of parametric patches. Sets of patches from the same category will follow certain constrained defined by templates. The templates are generated from some abstract representation of the category of interested (i.e. sets of cubbies).
The authors evaluate the method qualitatively on sketch-to-shape task, and show that the methods allows generating meshes that’s editable, interpretable, and with certain level of details.

Strength:
1. The representation of parametric patches have several advantages compared to prior works (whose patches are represented by neural networks). It can create sparse, compact, interpretable, and editable shapes that can be directly imported into many industrial design software.
2. The parametric patches allow computing CD without uniformly sampling from the surface.

Weakness:

1. The paper’s evaluation of the method is very limited. Even though the representation has many inherit advantages, I still expect the authors to demonstrate the method’s advantage in different tasks (not only sketch-modeling), different datasets (not just limited to rather small categories like Mugs), and across baselines with different representations (such as implicit representation, etc)
2.  A major weakness of the methods comes from the template generation. It occurs to me that the paper requires human user to pre-defined templates using a set of cuboids for each set of shapes before it can start generate. This makes it a little bit hard for the method to capture all topological variation of the category (which the author also mentioned in the last section). With this regard, the method requires additional information compared many prior methods (for example, implicit methods that can handle many topology and good details, and it doesn’t take a template).
3. There are potentially missing related works. Some of which I think the paper should compare with as baselines:

[A] Nash, Charlie, et al. "PolyGen: An autoregressive generative model of 3D meshes." arXiv preprint arXiv:2002.10880 (2020).
[B] Deng, Zhantao, et al. "Better Patch Stitching for Parametric Surface Reconstruction." arXiv preprint arXiv:2010.07021 (2020).

While the paper’s idea has its value as the representation could be useful for editing in many industrial software, but it requires more sophisticated evaluation (quantitative evaluation, more baselines, more tasks) in order to show-case the effectiveness of the representation. With that, I do not recommend the paper for ICLR.

---

> ### Author Response · Authors · 2020-11-11
> **Response**
>
> We thank the reviewer for their comments and suggestions. We are committed to improving our paper and can easily address all concerns within the rebuttal period. Below, we address each comment and indicate in **bold** additional experiments that we will post within several days. We hope that these will allay any concerns about our work and convince the reviewer that it will be a welcome contribution to the ICLR community. **If there are additional experiments that we can perform to further support our case, please let us know.**
>
> > I still expect the authors to demonstrate the method’s advantage in different tasks (not only sketch-modeling)
>
> Computing sparse and consistent patch-based representations of 3D shapes can benefit many modeling and design tasks. Beyond sketch-based modeling, single-view reconstruction, and interpolation, we will be glad to demonstrate additional applications. For instance, a classical problem in computer graphics is generating quad meshes or quad layouts for surfaces. Our method allows us to obtain a quad layout directly (by taking the boundary curves of our patches) and quad meshes trivially (by subdividing the layouts). **We will add an experiment that demonstrates this and compares to state-of-the-art procedural approaches.**
>
> The patch-based representation---largely unique to our work---is also a key feature motivated by practical considerations in computer graphics and computer-aided design. While some algorithms exist for converting other representations to this one, they can be unreliable and inefficient. We achieve comparable or improved quality relative to past work while bypassing this conversion step.
>
> > different datasets (not just limited to rather small categories like Mugs),
>
> We show results on a total of 8 ShapeNet categories, including airplanes (with over 4000 models) and cars (with over 7000 models).
>
> > It occurs to me that the paper requires human user to pre-defined templates using a set of cuboids for each set of shapes before it can start generate.
>
> We would like to clarify a misunderstanding: our method **does not** require manually constructed templates.
>
> While our system can use a custom template if desired, in Section 3.1 we introduce a purely data-driven algorithm for automatically generating a template for a shape category, without user guidance.
>
> Additionally, in Figure 5, we demonstrate all of our results with generic sphere templates. These templates are not shape- or category-specific but still produce high quality-results with only a small number (54) of patches.
>
> > There are potentially missing related works.
>
> We will add these citations to our text.
>
> Note that [A] generates meshes, which are not sparse and are more difficult to edit than our patch-based representations. Additionally, [A] is **not self-supervised**---it requires ground truth mesh data for training, whereas our method can be trained using any explicit surface representation, including point clouds.
>
> The reconstructions in [B] are obtained by optimizing the standard Chamfer distance and thus suffer from the sampling artifacts that our method addresses (see Section 3.2). Additionally, their patches are parameterized by neural networks (like in AtlasNet) and thus cannot be easily edited or rendered---even sampling points from each patch requires a forward pass through a deep network.
>
> Finally, we highlight that according to the ICLR guidelines, papers are considered contemporaneous if they are published within the last two months. [A] and [B] are not published works.

---

### Author Response · Authors · 2020-11-23
**Rebuttal revision**

We once again thank the reviewers for their helpful comments and suggestions. We have uploaded a revision that addresses all of the concerns raised in the reviews and subsequent discussion. In particular, we made the following changes:

### Qualitative results

We added many additional qualitative results for each of the eight shape categories in Sec. A.5. All of these models are rendered with patch boundaries so as to visualize the individual Coons patches.

### New applications

We added a section (Sec. 4.4) with two new experiments and comparisons to state-of-the art methods. In particular, we demonstrate how our system is able to convert 3D surfaces into **compact** and **consistent** NURBS patch decompositions. We compare our patches to those generated automatically by Rhino 3D, a popular commercial CAD tool. Not only does our method yield an order of magnitude fewer patches, but also our patches are consistent across different models. This makes our representation significantly more usable for further editing.

We reiterate that conversion of triangulated surfaces into NURBS patches is an open problem, and most solutions are extremely time-consuming and manual. In fact, the Rhino 3D documentation [1] states:

> This method converts each polygon face to a NURBS surface. It is not meant to convert entire mesh models to NURBS models and there is, in fact, no simple way to accomplish this.

There exist dedicated plug-ins to assist in such a conversion, but they require significant input from a trained user [2]. Our method produces a usable decomposition into patches automatically.

Additionally, in Sec 4.4, we show that a byproduct of our pipeline is a quad mesh of the input surface. Quad meshing is another popular computer graphics research problem, where successful quad meshes are those with few carefully-positioned singularities (vertices with valence not equal to four). By subdividing each of our predicted patches, we obtain a quad mesh whose number and position of singularities is entirely determined by the original template. Our resulting quad meshes have an order of magnitude fewer singularities than those generated using a modern specialized quad meshing algorithm [3].

### Additional ablation

We added additional qualitative examples to our ablation, as well as a quantitative ablation study on two shape categories in Sec. A.3.

### Minor improvements

We added a diagram of the control points and boundary curves that form a Coons patch to Fig. 3 for a more intuitive illustration of our representation.

We added the missing citations noted by R4 and R1.

We fixed the typos noted by R1.

We clarified our points regarding non-uniform sampling of Coons patches, as per the discussion with R3.

---

[1] https://developer.rhino3d.com/api/rhinoscript/mesh_methods/meshtonurb.htm

[2] https://youtu.be/ubsm0GfNnAQ

[3] Jakob et al. “Instant Field-Aligned Meshes.” SIGGRAPH Asia 2015.

---

### Decision · Program_Chairs · 2021-01-07
**Final Decision**

**Decision:**

Accept (Poster)

**Comment:**

Description:
The paper presents a patch-based 3D representation of man-made shapes that can be computed with deep learning and used directly in existing CAD applications. This representation is based off a deformable parametric template with Coons patches. Results in sketch-based modelling tasks shows comparable results with STOA

Strengths:
- The patch-based representation provide several advantages: compact, sparse, interpretable, consistent and easily editable.
- Can infer the right template, and thus does not require manually created templates

Weaknesses
- Limited evaluation restrained to mostly sketch-based modelling, and missing evaluation against a few STOA methods

The paper has introduced a very impactful new representation for 3D shapes and has strong technical novelty. I recommend, as reviewers have suggested, more in-depth quantitative evaluation (against other work and ablation studies)